# OpenReview forum: "VoiceAssistant-Eval: Benchmarking AI Assistants across Listening, Speaking, and Viewing"
_ICLR.cc/2026/Conference — ICLR 2026 Conference Withdrawn Submission_

### Official Review · Reviewer_yixd · 2025-10-23

**Soundness:** 3
**Presentation:** 4
**Contribution:** 3
**Rating:** 4
**Confidence:** 3

**Summary:**

This paper proposed a large benchmark for evaluating the speaking/listening/viewing abilities of voice assistant models. The proposed benchmark includes 10497 examples over 13 task categories. These tasks include listening(sounds, music, spoken dialogue...), role-play imitation, various scenarios for speaking, and images for viewing.  They evaluate 22 speech/omni models varied in different parameter sizes. The authors showed some insights from the results and conducted further error analysis on Qwen2.5-Omni-7B by humans, and also pointed out some challenges that remain in the field.

**Strengths:**

1. This is the first paper to include visual modality in speech assistant evaluation benchmark.
2. They included an exhaustive evaluation of recent SoTA speech/omni models on their benchmark.

**Weaknesses:**

1. The novelty of the benchmark is somewhat limited. Seems to me that most of the work done in this paper is merely merging existing data sources and using LLM and TTS tools to convert text questions into audio questions.
2. As an evaluation benchmark for daily use of voice assistants, I think the lack of real data is a major flaw of this benchmark.
3. The lack of human evaluation makes it hard for people to infer a voice assistant's performance in daily life from the results on this benchmark. For future researchers in this field, they cannot infer if these tasks are high-quality and reasonable for real-world voice assistant scenarios Therefore, it makes this benchmark less likely to be useful for the future research.

**Questions:**

1. For “Diverse Audio Contexts”, how did the authors choose the type of environmental sounds to overlap with the input speech?

---

> ### Author Response · Authors · 2025-11-18
> **Response to Official Review by Reviewer yixd (1/2)**
>
> **Q1:** The novelty of the benchmark is somewhat limited. Seems to me that most of the work done in this paper is merely merging existing data sources and using LLM and TTS tools to convert text questions into audio questions.
>
> **A1:** Thanks for your comments. We agree that this study is inspired by and builds upon existing research. However, we believe there is clear novelty and contribution in our tasks and metrics, which are the core components of our benchmark and deserve consideration. We discuss this in four aspects:
>
> First, **we are the first to evaluate models under hands-free, speech-only interaction with multi-modal (vision + audio) integration assessment**. We evaluate models’ speech outputs from multiple dimensions, including naturalness, emotion, and speaker tone. This cannot be achieved by evaluating performance across existing benchmarks, as they either lack sufficient multi-modal (vision + audio) integration or focus solely on text-based outputs.
>
> Second, we first organize the benchmarks for voice assistants into three main tasks: **listening**, **speaking**, and **viewing**. This structure more reasonably and comprehensively reflects the core abilities of voice assistants. For **all three tasks**, we use **speech as input** and evaluate the **speech output** of the models, which is a major distinction from existing benchmarks.
>
> Third, the ability to mimic a specific voice is crucial for creating personalized and engaging AI assistants. We are the **first** to evaluate models’ abilities in this aspect, which has not been included in existing benchmarks.
>
> Finally, as a benchmark work for the dataset and benchmark track, we carefully collected 47 existing datasets, reorganized them into 13 tasks under the three main categories, and synthesized speech input using three advanced TTS systems. **We also manually checked 15,359 evaluation samples, resulting in 10,497 high-quality samples**. This is clearly not a simple merge of existing benchmarks.
>
> **Q2:** As an evaluation benchmark for daily use of voice assistants, I think the lack of real data is a major flaw of this benchmark.
>
> **A2:** Thank you for your suggestion, and we apologize for not presenting this information clearly in the paper. We want to clarify some details:
>
> **First**, we would like to clarify our approach to audio instruction generation. As stated in the paper, "we first extract diverse visual and auditory inputs from 47 datasets" (line 120). For tasks originally presented with text instructions, such as listening tasks, viewing tasks, speaking_roleplay, and speaking_reasoning: "to replace the original text instructions, we synthesize speech instructions using three advanced text-to-speech (TTS) models" (line 124). This synthetic speech is used specifically to convert text-based instructions into audio format. However, recordings that already contain authentic audio from the source datasets are preserved. **We have added this to the main text: “Importantly, as detailed in Appendix E, our benchmark also includes 3,121 samples (29.7\%) with authentic human recordings from established corpora including Fisher, CommonVoice, LibriSpeech, MyST, and SpokenWOZ, ensuring our evaluation captures both synthetic and natural speech characteristics.”**
>
> Second, current state-of-the-art TTS models can generate speech that is perceptually indistinguishable from human speech. To validate this, we conduct a human study where we sample 1,500 user audio clips from our benchmark and ask three annotators (all unaware of the existence of this paper beforehand) to discriminate whether each audio clip was TTS-generated or human-recorded speech. The results show that only 10, 17, and 12 samples were identified as TTS-generated by them. These results indicate that **human annotators could not distinguish between synthetic and authentic speech, demonstrating that the TTS synthesis does not introduce artificial artifacts that would confound model evaluation**.
>
> **Finally**, we emphasize that every one of the **10,497 samples** in our final benchmark was manually verified from the initial **15,359 cleaned evaluation samples** to ensure the dataset’s overall quality and reliability.

---

> ### Author Response · Authors · 2025-11-18
> **Response to Official Review by Reviewer yixd (2/2)**
>
> **Q3:** The lack of human evaluation makes it hard for people to infer a voice assistant's performance in daily life from the results on this benchmark.
>
> **A3:** Thank you for your valuable suggestion.
>
> First, we apologize for any confusion: **we have included human evaluation, which was originally presented in Appendix B (Human Evaluation Validation) and has now been moved to the main text for better visibility**. Table 6 demonstrates **a very strong agreement between our evaluation scheme and human judgments** across various tasks. All reported Pearson correlation coefficients between the models' scores and human ratings are notably high (with an overall correlation of 0.92), indicating a strong linear relationship.
>
> Secondly, we emphasize that every one of the 10,497 samples in our final benchmark was manually verified from the initial 15,359 cleaned evaluation samples to ensure that these tasks are high-quality and reasonable for real-world voice assistant scenarios.
>
> **Q4:** For “Diverse Audio Contexts”, how did the authors choose the type of environmental sounds to overlap with the input speech?
>
> **A4:** Thank you for the question. First, we would like to clarify that our benchmark not only contains environmental sounds that overlap with the input speech. The “realistic conditions” in our context refer to natural sounds, music, and other complex auditory environments (lines 092–093). This corresponds mainly to our **Listening** subset. These questions are about the audio contexts, such as "What's the name of the music?" For speech under truly noisy backgrounds, those examples are primarily in the user_audios of the **speaking_robustness** task.
>
> **Second, regarding our taxonomy of audio types:** We identified four distinct categories based on acoustic characteristics and real-world usage patterns:
>
> 1.  **Music audio** encompasses various melodies, instrumental sounds, and songs
>
> 2.  **Speech audio** consists of human monologues or dialogues
>
> 3.  **Sound audio** includes general environmental sounds such as mechanical noises, animal calls, and natural sounds
>
> 4.  **General audio** represents complex mixtures combining elements from the above three categories
>
> This categorization allows us to systematically evaluate model performance across the full spectrum of auditory contexts that models encounter in real-world applications.

---

### Official Review · Reviewer_Lh6n · 2025-10-28

**Soundness:** 2
**Presentation:** 2
**Contribution:** 1
**Rating:** 2
**Confidence:** 4

**Summary:**

The paper claims that they introduces the first comprehensive benchmark designed to evaluate AI voice assistants across three modalities: listening, speaking, and viewing. The benchmark includes 10,497 curated examples covering 13 task categories.Extensive experiments shows that current models cannot do well in these task.

**Strengths:**

1. The paper is easy to follow.
2. The experiment covers a wide range of models.

**Weaknesses:**

1. The contribution of this work appears incremental. In the proposed benchmark, different tasks are disentangled and evaluated separately, which limits the motivation and novelty of the paper. Existing comprehensive benchmarks already provide suitable evaluation protocols. If the goal is to assess the capability of a universal speech model, this can be achieved by evaluating performance across multiple existing benchmarks.
2. The findings are intuitive. Claims such as “proprietary models cannot definitively surpass open-source models” are rather obvious. There is no model "definitely" surpass other models for all tasks. In fact, according to the experiments, proprietary models generally outperform open-source ones on average. The other two key findings are also trivial.
3. There are some existing audio-visual benchmark like [1][2][3], but there is no discussion in this paper. The claimed Weakness 4 regarding the limitations of existing benchmarks does not hold.
4. As a benchmark paper, the human evaluation step for assessing dataset quality, as well as the alignment between automatic evaluation and human judgment, is important. Please include these analyses in the main text rather than only in the appendix.
5. For role-playing tasks, models such as GPT-4o-Audio and Qwen2.5-Omni achieve nearly the highest scores, yet they use predefined speakers. What is the reason these models can still obtain such high scores?

I set my confidence score to 4 instead of 5 because I only flip through the appendices. According to the reviewer guidelines, we are not required to inspect the appendices in detail. After carefully reading the main text and briefly reviewing the appendices, I believe the motivation, novelty, and contribution of this paper do not meet the ICLR acceptance threshold. This is a fundamental issue that cannot be easily resolved by adding experiments or minor revisions, so I lean toward rejection and give a score of 2.

(If some important information is currently placed in the appendices, please let me know and revise the manuscript to move it into the main text. I would be happy to reconsider my score in that case.)

[1] Gong, Kaixiong, et al. "AV-Odyssey Bench: Can Your Multimodal LLMs Really Understand Audio-Visual Information?."

[2] Yao, Jihan, et al. "MMMG: a Comprehensive and Reliable Evaluation Suite for Multitask Multimodal Generation."

[3] Sung-Bin, Kim, et al. "Avhbench: A cross-modal hallucination benchmark for audio-visual large language models.

**Questions:**

1. Why is the UTMOS score threshold set to 3.8? I believe that for modern TTS systems, a UTMOS score of 3.8 does not represent high-quality audio.
2. For models that require a voice prompt, such as Step-Audio-2-mini, what kind of voice prompt do you use?

---

> ### Author Response · Authors · 2025-11-18
> **Response to Official Review by Reviewer Lh6n (1/3)**
>
> We sincerely thank you for your thoughtful feedback and careful review of our work. We address your concerns below:
>
> **Q1:** The contribution of this work appears incremental. In the proposed benchmark, different tasks are disentangled and evaluated separately, which limits the motivation and novelty of the paper. Existing comprehensive benchmarks already provide suitable evaluation protocols. If the goal is to assess the capability of a universal speech model, this can be achieved by evaluating performance across multiple existing benchmarks.
>
> **A1:** Thanks for your comments. We agree that this study is inspired by and builds upon existing research. However, we believe there is clear novelty and contribution in our tasks and metrics, which are the core components of our benchmark and deserve consideration. We discuss this in four aspects:
>
> First, **we are the first to evaluate models under hands-free, speech-only interaction with multi-modal (vision + audio) integration assessment**. We evaluate models’ speech outputs from multiple dimensions, including naturalness, emotion, and speaker tone. This cannot be achieved by evaluating performance across existing benchmarks, as they either lack sufficient multi-modal (vision + audio) integration or focus solely on text-based outputs. For example, we synthesize audio input based on the RoleBench and MMMU benchmarks.
>
> Second, regarding the concern that “different tasks are disentangled and evaluated separately,” we first organize the benchmarks for voice assistants into three main tasks: **listening**, **speaking**, and **viewing**. This structure more reasonably and comprehensively reflects the core abilities of voice assistants. For **all three tasks**, we use **speech as input** and evaluate the **speech output** of the models, which is a major distinction from existing benchmarks.
>
> Third, the ability to mimic a specific voice is crucial for creating personalized and engaging AI assistants. We are the **first** to evaluate models’ abilities in this aspect, which has not been included in existing benchmarks.
>
> Finally, as a benchmark work for the dataset and benchmark track, we carefully collected 47 existing datasets, reorganized them into 13 tasks under the three main categories, and synthesized speech input using three advanced TTS systems. **We also manually checked 15,359 evaluation samples, resulting in 10,497 high-quality samples**. This is not a simple merge of existing benchmarks.
>
> **Q2:** The findings are intuitive. Claims such as “proprietary models cannot definitively surpass open-source models” are rather obvious. There is no model that "definitely" surpasses other models for all tasks. In fact, according to the experiments, proprietary models generally outperform open-source ones on average. The other two key findings are also trivial.
>
> **A2:** We appreciate the reviewer's feedback and understand the concern about intuitive findings. To improve rigor, we have revised Finding 1 to state: “Open-source models can be highly competitive with proprietary models.” However, we respectfully suggest that **the value of our benchmark lies in transforming intuitions into concrete, quantified evidence that guides practical decision-making**.
>
> Regarding Finding 1, while no model universally excelling may seem self-evident, the critical contribution is identifying *which specific tasks* favor open-source models and *by what margin*. GPT-4o-Audio scores 9.1 points lower than Step-Audio-2-mini on Listening Speech, a substantial gap that challenges assumptions of proprietary superiority. **Without systematic evaluation, practitioners might default to expensive proprietary solutions where open-source alternatives actually perform better.**
>
> For Finding 2, systematic quantification reveals that most models perform better on speaking than listening tasks. This pattern identifies a clear bottleneck: audio encoders, not LLM backbones, primarily limit listening performance. The counter-example of Step-Audio-2-mini **demonstrates that this imbalance is addressable through targeted design choices**.
>
> For Finding 3, Step-Audio-2-mini (7B) achieves double the listening accuracy of LLaMA-Omni2-32B (40.06 vs 16.00). It demonstrates that specialized training and architectural choices can substitute for brute-force scaling, **which is critical for resource-constrained deployment scenarios.**
>
> Finally, we believe that rigorous benchmarks serve to validate or refute assumptions through reproducible evidence. While findings may appear "obvious" in retrospect, they require experimental proof to move from conjecture to established fact. **Our benchmark provides the quantitative foundation needed to guide future research and development in voice assistant systems.**

---

> ### Author Response · Authors · 2025-11-18
> **Response to Official Review by Reviewer Lh6n (2/3)**
>
> **Q3:** There are some existing audio-visual benchmarks, but there is no discussion in this paper. The claimed Weakness 4 regarding the limitations of existing benchmarks does not hold.
>
> **A3:** Thank you for raising this important point. We are aware that some audio-visual benchmarks exist in the literature. However, **these benchmarks predominantly focus on text-output evaluation, whereas our work addresses a fundamentally different paradigm: speech-output evaluation in voice assistant scenarios.** For example:
>
> -   “In AVQA, each question explicitly requires both the audio and visual signals from the clip to answer correctly.” **(line 1543)**
>
> -   “Questions in MUSIC-AVQA probe musical concepts (instrument identity, tempo changes, etc.) that often require both audio and visual understanding. Both AVQA and MUSIC-AVQA have been used to benchmark multimodal LLMs.” **(lines 1548–1550)**
>
> -   “A limitation is that these datasets emphasize short video clips (10–20 s) and specific domains (everyday scenes or music), so they cover only part of the audio–visual space.” **(line 1551)**
>
> **Regarding Weakness 4:** We acknowledge that our original phrasing was imprecise and may have suggested that no audio-visual benchmarks exist. This was not our intent. We have revised W4 to state more accurately: **"Insufficient multi-modal (vision+audio) integration assessment for voice assistants."**
>
> We apologize for overlooking some works as you mentioned. **We have clarified our statements and included these benchmarks in the Related Work section of the main text**: **AV-Odyssey** comprises 4,555 carefully curated problems that combine text, visual, and audio inputs, all formatted as multiple-choice questions. **OmniDialog** includes four dialogue types: text–text (1,455), visual–text (2,414), audio–text (883), and trimodal (165). The authors similarly convert these dialogues into multiple-choice questions. **AVHBench** is based on short videos and primarily evaluates models in terms of hallucination, audio–visual matching, and audio–visual captioning.
>
> These three works all **emphasize text-output understanding tasks**, whereas our benchmark introduces a voice-first, speech-output evaluation paradigm, representing a novel and complementary contribution to the field. Additionally, MMMG is a comprehensive benchmark for **multimodal generation** across four modality combinations (image, audio, interleaved text–image, and interleaved text–audio), but it only provides text input, addressing a different research direction.
>
> While these benchmarks are valuable, **they do not diminish the contribution of our dataset**. Our work complements existing efforts by specifically targeting speech-out, voice-first interaction scenarios.
>
> **Q4:** As a benchmark paper, the human evaluation step for assessing dataset quality, as well as the alignment between automatic evaluation and human judgment, is important. Please include these analyses in the main text rather than only in the appendix.
>
> **A4:** Thank you for your suggestion. For the human evaluation step for assessing dataset quality, we have added more details in Section 3.3 Data Curation and moved relevant information from Appendix J Data Curation Tool to the main text.
>
> For the alignment between automatic evaluation and human judgment, we have moved analyses in Appendix B Human Evaluation Validation to the main text.
>
> **Q5:** For role-playing tasks, models such as GPT-4o-Audio and Qwen2.5-Omni achieve nearly the highest scores, yet they use predefined speakers. What is the reason these models can still obtain such high scores?
>
> **A5:** Thank you for this insightful question. You are correct that GPT-4o-Audio achieves the highest overall score in role-playing tasks despite using predefined speakers. This can be attributed to the multi-dimensional nature of our evaluation framework.
>
> GPT-4o-Audio's superior performance stems primarily from its exceptionally strong content generation capability, backed by its powerful LLM backbone. **It achieves the highest Content score of 36.5**, significantly outperforming other models (e.g., Qwen2.5-Omni-7B scores only 12.7 in this dimension). This demonstrates that high-quality, contextually appropriate content generation is a critical component of role-playing performance.
>
> However, the use of predefined speakers does limit GPT-4o-Audio's performance in speaker-related metrics. Its Speaker Similarity score is 51.8, substantially lower than Step-Audio's 75.1. Similarly, while Qwen2.5-Omni-7B achieves the highest Consistency score (96.6) compared to Step-Audio's 90.5, it scores poorly in both Content (12.7) and Speaker Similarity (5.16), resulting in a much lower overall score of 5.2.

---

> > ### Comment · Reviewer_Lh6n · 2025-11-28
> > **Response to authors**
> >
> > I notice the authors claim that the novelty and motivation of this benchmark is that it is the first to assess voice assistants in a speech-in and speech-out setting. I recognize that evaluating such a setting is practical. However, is there any significant difference or insight the reader can get from this specific setting?
> >
> > Intuitively, performance should be comparable to or worse than the text-in and text-out setting, as the speech-in and speech-out task introduces more conditions and can be considered more difficult. Furthermore, for tasks involving paralinguistic information (e.g., tone or emotion), a text-input setting is not workable.

---

> > > ### Author Response · Authors · 2025-11-28
> > > **Thank you for your continued engagement**
> > >
> > > Dear Reviewer,
> > >
> > > Thank you for your continued engagement and thoughtful feedback. You raise an important point about the relationship between text-based and speech-based evaluation settings.
> > >
> > > We completely agree with your intuition that speech-in/speech-out evaluation represents a more challenging setting than text-in/text-out. As you correctly note, the speech modality introduces additional complexity through acoustic variability and the need to process paralinguistic information.
> > >
> > > Our empirical results strongly support this observation. Figure 3 demonstrates a substantial performance gap between text and speech inputs on identical questions. Specifically, all models exhibit significantly degraded performance when processing spoken instructions compared to text-based versions of the same content. **For instance, Qwen2.5-Omni-7B achieves 59.2% accuracy with text input but drops to 42.9% with audio input (a notable 16.3 percentage point decline).** This pattern holds consistently across all evaluated models, confirming that the speech setting is indeed more challenging.
> > >
> > > We appreciate you highlighting this distinction, as it reinforces the core motivation behind our benchmark.

---

> ### Author Response · Authors · 2025-11-18
> **Response to Official Review by Reviewer Lh6n (3/3)**
>
> **Q6:** Why is the UTMOS score threshold set to 3.8? I believe that for modern TTS systems, a UTMOS score of 3.8 does not represent high-quality audio.
>
> **A6:** Thank you for your suggestion, and we apologize for not presenting this information clearly in the paper. We want to emphasize that our benchmark aims to better reflect authentic human–assistant interactions and **should not only contain high-quality audio**.
>
> First, we would like to clarify that **varying audio quality among task types allows model performance to more accurately reflect their true capability in each domain**, rather than being influenced by unrelated factors. For example, speaking_reasoning yields the highest MOS (4.3) and viewing_multi_discipline follows closely (4.2), while speaking_robustness shows the lowest MOS (2.9), consistent with its challenging conditions.
>
> Second, we would like to clarify our approach to audio instruction generation. As stated in the paper, "we first extract diverse visual and auditory inputs from 47 datasets" (line 120). For tasks originally presented with text instructions, such as listening tasks and speaking_reasoning: "to replace the original text instructions, we synthesize speech instructions" (line 124). **We have added this to the main text: “Importantly, as detailed in Appendix E, our benchmark also includes 3,121 samples (29.7\%) with authentic human recordings from established corpora including Fisher, CommonVoice, LibriSpeech, MyST, and SpokenWOZ, ensuring our evaluation captures both synthetic and natural speech characteristics.”**
>
> Third, we emphasize that **every one of the 10,497 samples in our final benchmark was manually verified from the initial 15,359 cleaned evaluation samples to ensure the dataset's overall quality and reliability**.
>
> | Task                           | user_audio MOS |
> | ------------------------------ | -------------- |
> | listening_general              | 3.9            |
> | listening_music                | 4.1            |
> | listening_sound                | 3.9            |
> | listening_speech               | 4.0            |
> | speaking_assistant             | 4.2            |
> | speaking_emotion               | 4.1            |
> | speaking_instruction_following | 4.2            |
> | speaking_multi_round           | 3.6            |
> | speaking_reasoning             | 4.3            |
> | speaking_robustness            | 2.9            |
> | speaking_roleplay              | 4.1            |
> | speaking_safety                | 4.2            |
> | viewing_multi_discipline       | 4.2            |
>
> **Q7:** For models that require a voice prompt, such as Step-Audio-2-mini, what kind of voice prompt do you use?
>
> **A7:** Thank you for your question. For the role-playing task, we use the voice prompts provided in our benchmark. For other tasks, we use the default voice prompts provided in their official repositories. We have added this information to the experiment details section.

---

### Official Review · Reviewer_dwtY · 2025-10-31

**Soundness:** 3
**Presentation:** 2
**Contribution:** 3
**Rating:** 4
**Confidence:** 4

**Summary:**

This paper introduces **VoiceAssistant-Eval**, a comprehensive benchmark for evaluating multimodal AI assistants across *listening*, *speaking*, and *viewing* abilities. It includes over 10K curated samples and uses a triadic evaluation framework measuring content, speech, and consistency. Experiments on 22 models show that proprietary systems do not always outperform open-source ones, and that smaller, well-designed models can rival much larger ones. The benchmark is broad, systematic, and valuable for advancing voice-first AI evaluation. However, it relies entirely on **synthetic TTS audio**, which may limit realism, and lacks **barge-in (interruption) scenarios**, making it less fair for full-duplex systems like *Moshi* or *Freeze-Omni*. Incorporating real human speech and interaction-based tests would make it more representative of real-world usage.

**Strengths:**

- Comprehensive coverage – This is the first benchmark to simultaneously assess listening, speaking, and viewing in one unified framework, addressing several known limitations of prior voice or multimodal benchmarks.
- Systematic evaluation methodology – The triadic evaluation (content–speech–consistency) provides a holistic and interpretable view of model capabilities beyond text-only metrics.
- Broad and transparent comparison – Results across 22 models reveal key performance trends and highlight that smaller, well-aligned models can rival or exceed much larger ones.

**Weaknesses:**

- Lack of real human speech data. The benchmark relies entirely on TTS-synthesized audio for both prompts and responses. It is unclear whether performance on synthetic speech generalizes to real-world human recordings. I suggest incorporating real conversational corpora such as Fisher or similar datasets to better reflect authentic human–assistant interactions.

- Absence of interruption (barge-in) scenarios. The benchmark does not test situations where a user interrupts the assistant mid-speech, which are common in real human–AI conversations. This omission disadvantages full-duplex systems such as Moshi, Freeze-Omni and SALMONN-omni, which are explicitly designed to handle barge-in interactions. Even if most evaluated models are turn-based and cannot support such behavior, the benchmark should still include dedicated test cases to highlight the unique strengths of full-duplex voice assistants.

**Questions:**

- About Table 3: What are the specific characteristics and differences among the various tasks? It would be helpful if the authors could include one representative sample for each task in the appendix to illustrate its unique properties and how it differs from the others.


- Evaluation metrics across tasks: The specific evaluation metrics likely differ by subtask. In my opinion, emotion similarity should only be evaluated in the EMO subtask, and speaker similarity only in the RLP (role-play) subtask, is it right? Therefore, the detailed computation methods for each metric must vary across tasks. I suggest that the authors explicitly describe the evaluation criteria and metric formulations for each subtask in the appendix.

---

> ### Author Response · Authors · 2025-11-18
> **Response to Official Review by Reviewer dwtY (1/2)**
>
> **Q1:** Lack of real human speech data. The benchmark relies entirely on TTS-synthesized audio for both prompts and responses. It is unclear whether performance on synthetic speech generalizes to real-world human recordings. I suggest incorporating real conversational corpora such as Fisher or similar datasets to better reflect authentic human–assistant interactions.
>
> **A1:** Thank you for your suggestion, and we apologize for not presenting this information clearly in the paper. We want to clarify some details:
>
> **First**, we would like to clarify our approach to audio instruction generation. As stated in the paper, "we first extract diverse visual and auditory inputs from 47 datasets" (line 120). For tasks originally presented with text instructions, such as listening tasks and speaking_reasoning: "to replace the original text instructions, we synthesize speech instructions”. **Recordings that already contain authentic human audio from the source datasets are preserved.** We have added this to the main text: “Importantly, as detailed in Appendix E, **our benchmark also includes 3,121 samples (29.7\%) with authentic recordings from established corpora including Fisher, CommonVoice, LibriSpeech, MyST, and SpokenWOZ, ensuring our evaluation captures both synthetic and natural speech characteristics**.”
>
> **Second,** current state-of-the-art TTS models can generate speech that is perceptually indistinguishable from human speech. To validate this, we conduct a human study where we sample 1,500 user audio clips from our benchmark and ask three annotators (all unaware of the existence of this paper beforehand) to discriminate whether each audio clip was TTS-generated or human-recorded speech. The results show that only 10, 17, and 12 samples were identified as TTS-generated by them. These results indicate that **human annotators could not distinguish between synthetic and authentic speech**, demonstrating that the TTS synthesis does not introduce artificial artifacts that would confound model evaluation.
>
> **Finally**, we emphasize that every one of the **10,497 samples** in our final benchmark was manually verified from the initial **15,359 cleaned evaluation samples** to ensure the dataset’s overall quality and reliability.

---

> > ### Comment · Reviewer_dwtY · 2025-11-24
> >
> > Thank you for your response. However, I retain the following concerns:
> >
> > 1. Composition of specific subtasks: regarding your statement in the rebuttal: "For tasks originally presented with text instructions, such as listening tasks and speaking_reasoning: 'to replace the original text instructions, we synthesize speech instructions'." If I understand correctly, this implies that subtasks like listening tasks and speaking_reasoning are composed entirely of TTS-synthesized speech. Could you please report the specific ratio of real versus synthetic data for each individual subtask, rather than just the overall proportion for the entire benchmark?
> >
> > 2. Concerns regarding acoustic disparity and task difficulty: I am concerned that audio generated by TTS systems typically possesses higher sound quality and clarity compared to real human speech. Real-world recordings often contain background noise, reverberation, and non-standard pronunciation (or accents). I worry that relying exclusively on TTS data for certain tasks might inadvertently lower the difficulty level and fail to reflect real-world complexity.

---

> > > ### Author Response · Authors · 2025-11-24
> > > **Thank you for your continued engagement.**
> > >
> > > Thank you for your continued engagement. We appreciate the opportunity to provide more detailed clarification on these important points.
> > >
> > > **To Concern 1:**
> > >
> > > **Critical distinction:** We need to distinguish between two types of audio in our benchmark: user audio instructions (what the user says to the model) and context audio (the audio content being analyzed in listening tasks).
> > >
> > > **Example:** Given an audio clip of music playing with an unexpected sound interruption, the model is asked: "What unusual sound effect is heard in the song?" The audio clip containing the music and the unexpected sound is the context audio, while the spoken question that follows is the instruction.
> > >
> > > - For all listening tasks, we use TTS-synthesized user instructions, but the audio contexts are all real recordings. This design ensures models are tested on real-world acoustic phenomena while maintaining consistent instruction delivery.
> > >
> > > - For speaking tasks, the percentages are shown in the below table.
> > >
> > > | **Task**    | **speaking assistant** | **speaking emotion** | **speaking instruction_following** | **speaking multi_round** | **speaking reasoning** | **speaking robustness** | **speaking roleplay** | **speaking safety** |
> > > | ----------- | ---------------------- | -------------------- | ---------------------------------- | ------------------------ | ---------------------- | ----------------------- | --------------------- | ------------------- |
> > > | Real Speech | 82.8%                  | 80.4%                | 34.0%                              | 78.5%                    | 0                      | 89.6%                   | 0                     | 10.1%               |
> > >
> > > - The viewing task uses 0% real user instructions.
> > >
> > > **To Concern 2:**
> > >
> > > Your concern about TTS quality versus real-world complexity is well-founded. **We want to emphasize that as a comprehensive benchmark, our dataset covers 13 tasks and each task focuses on different abilities.** For background noise, reverberation, and non-standard pronunciation and accents that you mentioned, they are all included in the speaking robustness task, which is specially designed for this ability. **We have revised the paper to include detailed task descriptions for each task category in Appendix J.1 Overview of the 13 Evaluation Tasks.**
> > >
> > > In addition, we try to address your concern through multiple deliberate design choices that we detail below.
> > >
> > > Firstly, **all context audios in the listening tasks are real recordings**, and 44.9% of the speaking tasks have real human speech recordings.
> > >
> > > Secondly, **our UTMOS threshold of 3.8 is intentionally moderate, not representing pristine studio-quality audio**.
> > >
> > > Finally, **our human perceptual study provides strong evidence that modern TTS has achieved human-level realism**. Three independent annotators (not author) examined 1,500 audio samples from our benchmark. They were explicitly asked to identify which samples were TTS-generated versus authentic human recordings. The results showed that only 10, 17, and 12 samples were identified as synthetic by the three annotators respectively. This demonstrates that annotators could not distinguish between synthetic and authentic speech, providing empirical evidence that TTS artifacts do not bias our evaluation.

---

> ### Author Response · Authors · 2025-11-18
> **Response to Official Review by Reviewer dwtY (2/2)**
>
> **Q2:** Absence of interruption (barge-in) scenarios. The benchmark does not test situations where a user interrupts the assistant mid-speech, which are common in real human–AI conversations. This omission disadvantages full-duplex systems such as Moshi, Freeze-Omni and SALMONN-omni, which are explicitly designed to handle barge-in interactions. Even if most evaluated models are turn-based and cannot support such behavior, the benchmark should still include dedicated test cases to highlight the unique strengths of full-duplex voice assistants.
>
> **A2:** Thank you for raising this important point, and we apologize for this limitation in our current benchmark design. **We acknowledge this issue in Appendix A "Limitations and Future Work."** Specifically, on line 1120, we note: "VoiceAssistant-Eval evaluates models in an offline, static manner".
>
> We expand on this in lines 1122-1125: "Our current evaluation does not measure reaction time, the ability to handle partial utterances, or how well a model can manage barge-in (user interruption) and error recovery. Future work could explore an interactive evaluation framework, where models are tested in simulated real-time conversations or through user-in-the-loop studies to observe how they perform in a closed feedback loop."
>
> We sincerely agree that dedicated test cases for full-duplex capabilities would provide valuable insights into these systems' unique strengths. **However, implementing reliable barge-in testing would require a real-time interactive evaluation infrastructure that is significantly more complex and potentially less reproducible than our current offline approach. We are committed to exploring interactive evaluation frameworks in future work to better capture these dynamics and provide a more comprehensive assessment of both turn-based and full-duplex systems.**
>
> **Q3:** About Table 3: What are the specific characteristics and differences among the various tasks?
>
> **A3:** Thank you for this valuable suggestion. **We have revised the paper to include detailed task descriptions with representative examples for each task category in Appendix J.1 Overview of the 13 Evaluation Tasks.** Below is an example:
>
> **Listening General.** This task evaluates the model's ability to comprehend and respond to diverse everyday audio content, including conversations, environmental sounds, music, and ambient noise. The model must demonstrate robust audio processing capabilities across multiple acoustic contexts and interpret complex auditory scenes.
>
> **Representative example (Listening/General/8):** Given an audio clip of music playing with an unexpected sound interruption, the model is asked: "What unusual sound effect is heard in the song?" The correct response identifies the sound of glass shattering occurring during the musical playback, demonstrating the model's ability to isolate and recognize specific acoustic events within layered audio content.
>
> **Q4:** The specific evaluation metrics likely differ by subtask. In my opinion, emotion similarity should only be evaluated in the EMO subtask, and speaker similarity only in the RLP (role-play) subtask?
>
> **A4:** Thank you for your valuable suggention. All subtasks share the same evaluation metrics, with one exception, speaker similarity is only evaluated for role-play subtask. Different subtasks use task-specific evaluation prompts to guide the content quality assessment, but the underlying metric formulation remains consistent across tasks. **We have clarified the evaluation metrics more explicitly in Section 4:**
>
> The final score is computed as
>
> Final Score (%) = $ 100 \times (Q\_{\text{content}} \times Q\_{\text{speech}} \times Q\_{\text{consistency}} \times \max(\mathbb{I}[\text{task} \ne \text{roleplay}], Q\_{\text{speaker}}))$
>
> Content quality ($Q\_{\text{content}}$) measures informativeness, correctness, and emotional appropriateness via gpt‑oss‑20b, normalized to $[0,1]$. Speech quality ($Q\_{\text{speech}} = \text{UTMOS}/5 \in [0,1]$) quantifies perceptual naturalness and fluency using UTMOS. Cross‑modal consistency ($ Q\_{\text{consistency}} = 1 - \text{WER}'(text_1, text_2) $) evaluates alignment between generated transcriptions and original text. For role‑play tasks, speaker similarity ($Q\_{\text{speaker}}=\cos(e\_{gen}, e\_{ref})$) measures voice resemblance using Wespeaker embeddings.
>
> The emotion prompts serve to evaluate models' ability to provide emotionally appropriate responses when users need emotional support or empathy. We use an emotion vector model to extract and quantify the emotions expressed in model outputs and provide these results to gpt‑oss‑20b, which determines whether the emotion is appropriate. This assessment is incorporated into the content quality score ($Q\_{\text{content}}$), ensuring that responses demonstrate genuine affective alignment with user needs.
>
> We demonstrate our evaluation prompts for each subtask in Appendix K Experiment Details.

---

### Official Review · Reviewer_VieU · 2025-10-31

**Soundness:** 3
**Presentation:** 1
**Contribution:** 3
**Rating:** 4
**Confidence:** 4

**Summary:**

This paper presents VoiceAssistant-Eval, a comprehensive benchmark for evaluating AI assistants across listening, speaking, and viewing capabilities. It consolidates 13 diverse tasks spanning speech understanding, speech generation, multimodal grounding, and role-play imitation, and introduces a triadic evaluation protocol that jointly measures response consistency, speech quality, and content quality. Experiments on 21 open-source and proprietary systems reveal distinctive strengths and weaknesses among models and offer insightful directions for future research.

**Strengths:**

* Addresses a timely and important research problem by focusing on the speech-in/speech-out evaluation paradigm, an emerging yet underexplored direction in developing voice assistants and omni-modal LLMs.
* Provides clear and well-motivated problem framing, effectively articulating the limitations of current benchmarks (W1-W4) and demonstrating how these gaps motivate the proposed benchmark design.

**Weaknesses:**

* The paper critically fails to provide clear definitions for each of the 13 tasks. For example, "speaking-robustness" is indeed ambiguous. Does it test resistance to input noise, adversarial prompts, or something else? This lack of clarity undermines the benchmark's representativeness and positioning and makes it impossible to assess whether the tasks actually measure what they claim to measure.
* The technical novelty of the paper is limited, as its primary contribution lies in dataset aggregation and curation rather than methodological innovation. The benchmark primarily integrates 47 existing datasets with synthetic speech and applies standard evaluation metrics.
* Many tasks, especially for those under the speaking and viewing categories, appear to emphasize surface-level semantic understanding of speech before solving the task itself. This raises an important question: how much performance difference arises from using audio input versus text input? A baseline system employing a text-based variant (to isolate modality effects) and a cascade pipeline (ASR --> LLM/VLLM --> TTS) could serve as a strong comparison point, yet such baselines are not included in the current study.
* Related to the previous one. Several critical aspects of speech-centric understanding that are fundamental to voice assistants—such as reasoning about intonation, prosody, and pause timing to infer intention, or conditioning responses on speaker emotion and stress—are not considered. Moreover, robustness to noise and acoustic variation is not explicitly evaluate and discussed. A discussion of these limitations and possible extensions would strengthen the impact of this benchmark.
* Using a speaker embedding model as a primary evaluation tool for the role-play task risks overemphasizing timbre similarity rather than capturing prosodic style, speaking tone, or persona-specific delivery.
* The claim that current models perform better on speaking tasks than on listening tasks is also problematic, since the relative difficulty levels for humans across these categories are unknown. Without a human performance baseline or a normalization for task difficulty, such cross-category comparisons risk being non-equivalent and may not reflect genuine model limitations.

**Questions:**

* Given the paper’s focus on voice assistants, it would be advisable to include additional proprietary baselines (e.g. Gemini) in the comparative evaluation.
* The paper should clearly specify how the triadic score is aggregated (e.g., geometric mean vs. weighted sum) and provide a clear justification for this design choice.
* It would be helpful to clarify whether and how the leaderboard will be made publicly available.
* A majority of the identified weaknesses stem from unclear presentation, insufficient justification of design choices, and limited discussion of potential limitations. Addressing these issues would substantially strengthen the paper and could justify a higher review score upon revision.

---

> ### Author Response · Authors · 2025-11-18
> **Response to Official Review by Reviewer VieU (1/3)**
>
> **Q1:** The paper critically fails to provide clear definitions for each of the 13 tasks. For example, "speaking-robustness" is indeed ambiguous. Does it test resistance to input noise, adversarial prompts, or something else?
>
> **A1:** Thank you for this valuable suggestion. **We have revised the paper to include detailed task descriptions with representative examples for each task category in Appendix J.1 Overview of the 13 Evaluation Tasks.** Below are two illustrative examples:
>
> **Speaking Robustness.** This task assesses the model's resilience and reliability when handling challenging audio conditions, such as background noise, poor audio quality, diverse accents, speech impediments, or ambiguous input. It evaluates how well the model maintains performance under suboptimal circumstances, accurately processes degraded audio signals, and recovers from recognition difficulties. The model must demonstrate stability, adaptability, and consistent functionality across varied and adverse acoustic environments.
>
> **Speaking Safety.** This task assesses the model's adherence to safety guidelines and ethical standards in voice interactions. It evaluates how well the model identifies and handles potentially harmful requests, recognizes requests for offensive or discriminatory content, avoids generating dangerous or unethical material, and maintains appropriate boundaries. The model must demonstrate responsible behavior by clearly refusing unsafe requests, explaining why such requests are inappropriate when helpful, providing constructive alternatives when possible, and protecting user wellbeing while remaining respectful and accessible.
>
> **Q2:** The technical novelty of the paper is limited, as its primary contribution lies in dataset aggregation and curation rather than methodological innovation. The benchmark primarily integrates 47 existing datasets with synthetic speech and applies standard evaluation metrics.
>
> **A2:** Thanks for your comments. We agree that this study is inspired by and builds upon existing research. However, we believe there is clear novelty and contribution in our tasks and metrics, which are the core components of our benchmark and deserve consideration. We discuss this in four aspects:
>
> First, **we are the first to evaluate models under hands-free, speech-only interaction with multi-modal (vision + audio) integration assessment**. We evaluate models’ speech outputs from multiple dimensions, including naturalness, emotion, and speaker tone. This cannot be achieved by evaluating performance across existing benchmarks, as they either lack sufficient multi-modal (vision + audio) integration or focus solely on text-based outputs. For example, we synthesize audio input based on the RoleBench and MMMU benchmarks.
>
> Second, we first organize the benchmarks for voice assistants into three main tasks: **listening**, **speaking**, and **viewing**. This structure more reasonably and comprehensively reflects the core abilities of voice assistants. For **all three tasks**, we use **speech as input** and evaluate the **speech output** of the models, which is a major distinction from existing benchmarks.
>
> Third, the ability to mimic a specific voice is crucial for creating personalized and engaging AI assistants. We are the **first** to evaluate models’ abilities in this aspect, which has not been included in existing benchmarks.
>
> Finally, as a benchmark work for the dataset and benchmark track, we carefully collected 47 existing datasets, reorganized them into 13 tasks under the three main categories, and synthesized speech input using three advanced TTS systems. **We also manually checked 15,359 evaluation samples, resulting in 10,497 high-quality samples**. This is not a simple merge of existing benchmarks.
>
> **Q3:** How much performance difference arises from using audio input versus text input?
>
> **A3:** Thank you for this important question. We actually provide a text-based variant of the viewing task, and the results are shown in Figure 3. All evaluated models perform substantially worse when queries are spoken rather than written, illustrating the current gap in robust audio-visual integration.

---

> ### Author Response · Authors · 2025-11-18
> **Response to Official Review by Reviewer VieU (2/3)**
>
> **Q4:** Several critical aspects of speech-centric such as reasoning about intonation, prosody, and pause timing to infer intention, or conditioning responses on speaker emotion and stress are not considered. Moreover, robustness to noise and acoustic variation is not explicitly evaluated and discussed. A discussion of these limitations and possible extensions would strengthen the impact of this benchmark.
>
> **A4:** Thank you for this valuable suggestion. **We discuss these limitations in Appendix A "Limitations and Future Work."** On line 1120, we note: "Finally, VoiceAssistant-Eval evaluates models in an offline, static manner, which might not capture certain practical deployment factors." Regarding acoustic robustness, **we do evaluate noise and acoustic variation through our Speaking Robustness task**, which tests model performance under various acoustic conditions. Lines 1122-1125 state: "Our current evaluation does not measure reaction time, the ability to handle partial utterances, or how well a model can manage barge-in (user interruption) and error recovery. Future work could explore an interactive evaluation framework, where models are tested in simulated real-time conversations or through user-in-the-loop studies to observe how they perform in a closed feedback loop."
>
> **Q5:** Using a speaker embedding model as a primary evaluation tool for the role-play task risks overemphasizing timbre similarity rather than capturing prosodic style, speaking tone, or persona-specific delivery.
>
> **A5:** Thank you for your comment. We would like to clarify that the speaker embedding model is not the primary evaluation tool, but rather one of four evaluation dimensions. The other three dimensions are speech quality, content quality, and speech-content consistency. We use gpt‑oss‑20b to persona-specific delivery aspects such as prosodic style, speaking tone, and character-appropriate expression.
>
> **Q6:** The claim that current models perform better on speaking tasks than on listening tasks is also problematic, since the relative difficulty levels for humans across these categories are unknown. Without a human performance baseline or a normalization for task difficulty, such cross-category comparisons risk being non-equivalent and may not reflect genuine model limitations.
>
> **A6:** Thank you for this observation. The speaking and listening scores are directly comparable because both use **identical input-output formats (audio-to-audio) and the same evaluation framework**.
>
> The consistent pattern (20 of 22 models scoring higher on speaking than listening) is what makes this finding meaningful. While both involve audio processing, they test fundamentally different capabilities: speaking assesses generation while listening assesses comprehension. This systematic asymmetry suggests important differences in how current architectures handle these distinct challenges. The exceptions validate our comparison. Step-Audio-2-mini scores 40.06 on listening versus 31.30 on speaking, demonstrating that our framework successfully distinguishes between these capabilities.
>
> You're right that without human baselines, we cannot claim absolute task difficulty. However, **our findings still reveal relative strengths and weaknesses across model architectures, which is valuable for understanding the field and informing future development.**
>
> **Q7:** Given the paper’s focus on voice assistants, it would be advisable to include additional proprietary baselines (e.g. Gemini) in the comparative evaluation.
>
> **A7:** Thank you for this valuable observation. Following your suggestion, we have evaluated Gemini 2.5 Flash (released September 18, 2025), the latest Gemini model offering audio output support, alongside the open-source Qwen3-Omni-30B-A3B-Instruct (also released in September 2025).
>
> The results show that Qwen3-Omni outperforms Gemini 2.5 Flash in both listening and speaking tasks, while Gemini demonstrates superior performance on viewing tasks. These findings reinforce our central conclusion that open-source models can achieve competitive, and sometimes superior, performance compared to proprietary alternatives across different modalities. This updated comparison with the latest available proprietary model thus strengthens our original findings.
>
> | Model                       | Time    | Listening | Speaking | Viewing |
> | --------------------------- | ------- | --------- | -------- | ------- |
> | Qwen2.5-Omni-7B             | 2025-07 | 33.6      | 41.3     | 34.3    |
> | GPT-4o-Audio                | 2025-06 | 39.8      | 51.3     | --      |
> | Qwen3-Omni-30B-A3B-Instruct | 2025-09 | 45.6      | 53.3     | 34.5    |
> | Gemini-2.5-flash            | 2025-09 | 34.3      | 45.2     | 46.2    |

---

> ### Author Response · Authors · 2025-11-18
> **Response to Official Review by Reviewer VieU (3/3)**
>
> **Q8:** The paper should clearly specify how the triadic score is aggregated (e.g., geometric mean vs. weighted sum) and provide a clear justification for this design choice.
>
> **A8:** We have added the below formula in Section 4:
>
> The final score is computed as
>
> Final Score (%) = $ 100 \times (Q\_{\text{content}} \times Q\_{\text{speech}} \times Q\_{\text{consistency}} \times \max(\mathbb{I}[\text{task} \ne \text{roleplay}], Q\_{\text{speaker}}))$
>
> Content quality ($Q\_{\text{content}}$) measures informativeness, correctness, and emotional appropriateness via gpt‑oss‑20b, normalized to $[0,1]$. Speech quality ($Q\_{\text{speech}} = \text{UTMOS}/5 \in [0,1]$) quantifies perceptual naturalness and fluency using UTMOS. Cross‑modal consistency ($ Q\_{\text{consistency}} = 1 - \text{WER}'(text_1, text_2) $) evaluates alignment between generated transcriptions and original text. For role‑play tasks, speaker similarity ($Q\_{\text{speaker}}=\cos(e\_{gen}, e\_{ref})$) measures voice resemblance using Wespeaker embeddings.
>
> **This approach is deliberately chosen, as it effectively highlights even minor deficiencies across multiple aspects of model performance**. Any weakness in one dimension ultimately limits the end-to-end user experience, and these components are not substitutable. A high-quality text output cannot compensate for poor acoustic rendering, and a perfect waveform cannot compensate for incorrect or misaligned content. Our aim with VoiceAssistant-Eval is to push the boundaries of what AI assistants can achieve. We believe it will be instrumental in developing next-generation multimodal models and in tracking progress toward truly expert AI assistants.
>
> To illustrate this design choice: Consider a model achieving near-perfect content quality (0.95) and speech quality (0.95), but severe text-to-speech misalignment (0.20). An averaging approach would yield a score around 0.70, suggesting acceptable performance. However, the multiplicative approach yields 0.18, which better reflects reality: a voice assistant with frequent speech-content misalignment is fundamentally unusable, regardless of its other strengths. The multiplicative scoring captures the practical constraint that all dimensions must meet minimum quality thresholds for system viability.
>
> **Q9:** It would be helpful to clarify whether and how the leaderboard will be made publicly available.
>
> **A9:** A project page that provides the leaderboard will be maintained. We will provide both the main leaderboard (with the multiplicative step) and sub-leaderboards showing individual scores, as presented in Appendix G: Detailed Assessment of Model Capabilities.

---

### Official Review · Reviewer_AjBh · 2025-11-01

**Soundness:** 3
**Presentation:** 1
**Contribution:** 3
**Rating:** 6
**Confidence:** 3

**Summary:**

This paper introduce VoiceAsssistant-Eval, a comprehensive benchmark designed to assess AI assistants across listening, speaking, and viewing, comprising 10,497 curated examples spanning 13 task categories. The evaluation focus on hands-free interaction and "voice-first". The measurement aspects include the quality of response content, speech, and their consistency. They evaluated 21 open-source models and GPT-4o-Audio, and reveal some key findings.

**Strengths:**

1. Closer to real voice assistant evaluation: in practical scenarios, voice assistants need to support speech-only interaction. In this benchmark, all examples use speech rather than text for interaction (using speech as input and output). In the evaluation of listening, in addition to speech, it also covers everyday soundscapes including natural sounds, music, environmental noises.

2. Comprehensive test categories: including four listening tasks (general, music, sound, speech), eight speaking tasks (assistant, emotion, instruction following, multi-round, reasoning, robustness, role playing, safety), and one viewing task (multi-discipline). The sample numbers among different categories are basically balanced. They evaluate across three dimensions: content quality, speech quality, and consistency between them.

3. A comprehensive evaluation and comparison on 21 open-source models and one closed-source model GPT-4o-Audio, with findings: (1) Proprietary models cannot definitively surpass open-source models; (2) Current models tend to perform better on speaking tasks than on listening tasks; (3) Smaller, well-architected models can rival larger models; (4) Role-play tasks are challenging; (5) Multi-modal (vision+audio) integration remains a challenge; (6) Safety alignment and robustness still require further improvement.

4. Authors conducted a consistency check between gpt-oss-20b and human evaluation, showing that gpt-oss-20b align with human preference.

**Weaknesses:**

1. As you said, one weakness of other benchmarks is “they rarely evaluate models under realistic conditions with varied audio contexts.” (line 090-091). However, in your speech synthesis, you filter with UTMOS and regenerate until the score > 3.8, then pick the lowest-WER sample via Whisper. This pipeline likely biases toward clean, high-MOS TTS and away from truly noisy/realistic conditions (real-life recordings with UTMOS<3 are common). How do your benchmark avoid the issue of “don't evaluated under realistic conditions”?

2. Overclaim in Related Work section: line 181 mentions that previous benchmarks did not have audio/visual context or voice imitation; line 191 claims that VoiceAssistant-Eval is first to evaluate models in rich audio–visual contexts. However, OmniDialog[1] (trimodal text-vision-audio dialogues) and AV-Odyssey Bench[2] (4,555 multimodal problems integrating text, visual and audio) already evaluate audio-visual (+text) understanding.  I acknowledge your benchmark’s speech-out, voice-first setting is novel and valuable, but saying previous benchmarks lacked audio/visual context is incorrect.

3. Aggregating multiple metrics into a single score will harm the fine-grained evaluation of content quality and speech quality. As shown in Appendix G, there is a trade-off between content and speech quality. Aggregating them into a single score will lead to the inability to evaluate a model’s speech and content capabilities separately.

4. The presentation of the Triadic Evaluation System is very hard to follow. Important details should be in the main text with detailed explanation or formula.

5. The findings about proprietary vs. open-source models may be outdated, as GPT-4o-Audio is from early 2024. Including more recent proprietary models would strengthen the comparison.

[1] Razzhigaev, Anton, et al. "OmniDialog: A Multimodal Benchmark for Generalization Across Text, Visual, and Audio Modalities." Proceedings of the 2nd GenBench Workshop on Generalisation (Benchmarking) in NLP. 2024.

[2] Gong, Kaixiong, et al. "AV-Odyssey Bench: Can Your Multimodal LLMs Really Understand Audio-Visual Information?." arXiv preprint arXiv:2412.02611 (2024).

**Questions:**

1. Why do you think the ability of role playing is important? Would model's affective expressivity and empathy (without mimicking a specific speaker) be a more direct driver of user trust and sustained engagement than timbre imitation?

2. About the triadic evaluation system section, how are the three individual scores calculated respectively? What is the purpose of emotion prompts, and why keep emotions with probability >1% (not another number, for example, 10%)? How is the length threshold WER for multiple-choice questions done? What do the formulas in lines 301–307 mean? Why choose multiplication of three subscores rather than an additive/geometric aggregation? Why are the speaking score and listening score comparable (line 368, "20 out of 22 models score higher on Speaking than on Listening")?

3. In your role playing task, do you consider different languages, accents, and genders?

---

> ### Author Response · Authors · 2025-11-18
> **Response to Official Review by Reviewer AjBh (1/4)**
>
> **Q1:** In your speech synthesis, you filter with UTMOS and regenerate until the score \> 3.8, then pick the lowest-WER sample via Whisper. This pipeline likely biases toward clean, high-MOS TTS and away from truly noisy/realistic conditions (real-life recordings with UTMOS\<3 are common). How do your benchmark avoid the issue of “don't evaluated under realistic conditions”?
>
> **A1:** Thank you for your suggestion, and we apologize for not presenting this information clearly in the paper. We want to clarify some details:
>
> **First**, we would like to clarify that “realistic conditions” in our context refer to *natural sounds, music, and other complex auditory environments* (lines 092–093). This corresponds mainly to our **Listening** subset, and these questions are about the audio contexts, such as "What's the name of the music?" For speech under truly noisy backgrounds, those examples are primarily in the *sound_audio* of the **listening_speech** task and the *user_audios* of the **speaking_robustness** task. As shown in the below table, **speaking_robustness** samples have a MOS of *2.9*, and **sound_audio** in **listening_speech** has a MOS of *2.6*. It is worth noting that sound_audio from other listening tasks consists mostly of non-speech sounds, so their MOS values are typically below *2.0*.
>
> **Second**, we would like to clarify our approach to audio instruction generation. As stated in the paper, "we first extract diverse visual and auditory inputs from 47 datasets" (line 120). For tasks originally presented with text instructions, such as listening tasks and speaking_reasoning: "to replace the original text instructions, we synthesize speech instructions”. Recordings that already contain authentic human audio from the source datasets are preserved. **We have added this to Section 3.2: “Importantly, as detailed in Appendix E, our benchmark also includes 3,121 samples (29.7\%) with authentic human recordings from established corpora including Fisher, CommonVoice, LibriSpeech, MyST, and SpokenWOZ, ensuring our evaluation captures both synthetic and natural speech characteristics.”**
>
> **Third**, we would like to clarify that varying audio quality among task types allows model performance to more accurately reflect their true capability in each domain, rather than being influenced by unrelated factors. For example, **speaking_reasoning** yields the highest MOS (4.3) and **viewing_multi_discipline** follows closely (4.2), while **speaking_robustness** shows the lowest MOS (2.9), consistent with its challenging conditions.
>
> **Finally**, we emphasize that every one of the **10,497 samples** in our final benchmark was manually verified from the initial 15,359 cleaned evaluation samples to ensure the dataset’s overall quality and reliability.
>
> | Task                           | user_audio MOS | sound_audio MOS |
> | ------------------------------ | -------------- | --------------- |
> | listening_general              | 3.9            | 1.6             |
> | listening_music                | 4.1            | 1.4             |
> | listening_sound                | 3.9            | 1.4             |
> | listening_speech               | 4.0            | 2.6             |
> | speaking_assistant             | 4.2            | \-              |
> | speaking_emotion               | 4.1            | \-              |
> | speaking_instruction_following | 4.2            | \-              |
> | speaking_multi_round           | 3.6            | \-              |
> | speaking_reasoning             | 4.3            | \-              |
> | speaking_robustness            | 2.9            | \-              |
> | speaking_roleplay              | 4.1            | \-              |
> | speaking_safety                | 4.2            | \-              |
> | viewing_multi_discipline       | 4.2            | \-              |

---

> ### Author Response · Authors · 2025-11-18
> **Response to Official Review by Reviewer AjBh (2/4)**
>
> **Q2:** Overclaim in Related Work: line 181 mentions that previous benchmarks did not have audio/visual context or voice imitation; line 191 claims that VoiceAssistant-Eval is first to evaluate models in rich audio–visual contexts. However, OmniDialog[1] and AV-Odyssey Bench[2] already evaluate audio-visual (+text) understanding. I acknowledge your benchmark’s speech-out, voice-first setting is novel and valuable, but saying previous benchmarks lacked audio/visual context is incorrect.
>
> **A2:** Thank you for pointing out the omission of the works you mentioned.
>
> **Firstly**, we would like to clarify that in line 161 (the line before 181), the phrase “All of these benchmarks” refers specifically to the five *Speech Interaction Benchmarks* discussed earlier. **We have changed it to “All of these five benchmarks”.**
>
> **Secondly**, actually we have surveyed some **benchmarks that have audio/visual context, but they all rely on text-only output and have limitations on their coverage and scale:**
>
> -   “In AVQA, each question explicitly requires both the audio and visual signals from the clip to answer correctly.” (line 1543)
>
> -   “Questions in MUSIC-AVQA probe musical concepts (instrument identity, tempo changes, etc.) that often require both audio and visual understanding. Both AVQA and MUSIC-AVQA have been used to benchmark multimodal LLMs.” (lines 1548–1550)
>
> -   “A limitation is that these datasets emphasize short video clips (10–20 s) and specific domains (everyday scenes or music), so they cover only part of the audio–visual space.” (line 1551)
>
> So in line 191, where we say our benchmark is the first to evaluate models in rich audio–visual contexts, we have changed it to: "our benchmark differs by evaluating models in richer audio–visual contexts within voice-first, speech-output interaction settings."
>
> **Thirdly**, we apologize for overlooking some works as you mentioned. We have clarified our statements and included these benchmarks in the *Related Work* section of the main text:
>
> **AV-Odyssey** comprises 4,555 carefully curated problems that combine text, visual, and audio inputs, all formatted as multiple-choice questions. **OmniDialog** includes four dialogue types: text–text (1,455), visual–text (2,414), audio–text (883), and trimodal (165). The authors similarly convert these dialogues into multiple-choice questions. **AVHBench** is based on short videos and primarily evaluates models in terms of hallucination, audio–visual matching, and audio–visual captioning.
>
> These three works all **emphasize text-output understanding tasks**, whereas our benchmark introduces a voice-first, speech-output evaluation paradigm, representing a novel and complementary contribution to the field. Additionally, MMMG is a comprehensive benchmark for **multimodal generation** across four modality combinations (image, audio, interleaved text–image, and interleaved text–audio), but it only provides text input, addressing a different research direction.
>
> While these benchmarks are valuable, **they do not diminish the contribution of our dataset**. Our work complements existing efforts by specifically targeting speech-out, voice-first interaction scenarios.
>
> **Q3:** Why choose multiplication of three subscores rather than an additive/geometric aggregation?
>
> **A3:** Thank you for your suggestion. We acknowledge that the multiplicative scoring approach may excessively penalize a model for weaker performance in any single dimension. However, **this approach is deliberately chosen, as it effectively highlights even minor deficiencies across multiple aspects of model performance**. Any weakness in one dimension ultimately limits the end-to-end user experience, and these components are not substitutable. A high-quality text output cannot compensate for poor acoustic rendering, and a perfect waveform cannot compensate for incorrect or misaligned content. Our aim with VoiceAssistant-Eval is to push the boundaries of what AI assistants can achieve. We believe it will be instrumental in developing next-generation multimodal models and in tracking progress toward truly expert AI assistants.
>
> To illustrate this design choice: Consider a model achieving near-perfect content quality (0.95) and speech quality (0.95), but severe text-to-speech misalignment (0.20). An averaging approach would yield a score around 0.70, suggesting acceptable performance. However, the multiplicative approach yields 0.18, which better reflects reality: a voice assistant with frequent speech-content misalignment is fundamentally unusable, regardless of its other strengths. The multiplicative scoring captures the practical constraint that all dimensions must meet minimum quality thresholds for system viability.
>
> In addition, **we will maintain both the main leaderboard (with the multiplicative score) for advanced general models and sub-leaderboards showing individual scores for task-specific models**.

---

> ### Author Response · Authors · 2025-11-18
> **Response to Official Review by Reviewer AjBh (3/4)**
>
> **Q4:** The findings about proprietary vs. open-source models may be outdated, as GPT-4o-Audio is from early 2024.
>
> **A4:** Thank you for this valuable observation. Following your suggestion, we have evaluated Gemini 2.5 Flash (released September 18, 2025), the latest Gemini model offering audio output support, alongside the open-source Qwen3-Omni-30B-A3B-Instruct (also released in September 2025).
>
> The results show that Qwen3-Omni outperforms Gemini 2.5 Flash in both listening and speaking tasks, while Gemini demonstrates superior performance on viewing tasks. These findings reinforce our conclusion that open-source models can achieve competitive, and sometimes superior, performance compared to proprietary alternatives across different modalities.
>
> | Model                       | Time    | Listening | Speaking | Viewing |
> | --------------------------- | ------- | --------- | -------- | ------- |
> | Qwen2.5-Omni-7B             | 2025-07 | 33.6      | 41.3     | 34.3    |
> | GPT-4o-Audio                | 2025-06 | 39.8      | 51.3     | --      |
> | Qwen3-Omni-30B-A3B-Instruct | 2025-09 | 45.6      | 53.3     | 34.5    |
> | Gemini-2.5-flash            | 2025-09 | 34.3      | 45.2     | 46.2    |
>
> **Q5:** Why do you think the ability of role playing is important? Would model's affective expressivity and empathy be a more direct driver of user trust and sustained engagement than timbre imitation?
>
> **A5:** We appreciate this thoughtful question. You're right that affective expressivity and empathy are crucial drivers of user trust and engagement. Our benchmark actually evaluates both dimensions. The role-playing metric assesses not just timbre imitation, but a holistic combination of factors:
>
> $\text{Final Score}$ (%) = $100 \times (Q\_{\text{content}} \times Q\_{\text{speech}} \times Q\_{\text{consistency}} \times Q\_{\text{speaker}})$
>
> Additionally, our benchmark explicitly evaluates affective expressivity through the "speaking assistant" task and empathy through the "speaking emotion" task, treating these as essential capabilities alongside role-playing.
>
> Actually, **role-playing represents a frontier research direction as evidenced by recent publications**:
>
> -   Personalized Visual Instruction Tuning (ICLR 2025)
>
> -   MMRole: A Comprehensive Framework for Developing and Evaluating Multimodal Role-Playing Agents (ICLR 2025)
>
> -   Aligning VLM Assistants with Personalized Situated Cognition (ACL 2025)
>
> -   Visual Persona: Foundation Model for Full-Body Human Customization (CVPR 2025)
>
> -   From Persona to Personalization: A Survey on Role-Playing Language Agents (TMLR 2024)
>
> However, these existing works overlook a critical dimension: speaker timbre. For users, voice characteristics are deeply intertwined with identity and character recognition. A character who sounds completely different from their established voice breaks immersion just as much as inconsistent personality traits would. By including timbre fidelity in evaluation, we provide a more complete assessment of multimodal role-playing capabilities that aligns with how humans actually experience character interactions.
>
> **Q6:** About the triadic evaluation system section, how are the three individual scores calculated respectively?
>
> **A6:** Sorry for the insufficient description. We have added the below in the main text:
>
> The final score is computed as
>
> Final Score (%) = $ 100 \times (Q\_{\text{content}} \times Q\_{\text{speech}} \times Q\_{\text{consistency}} \times \max(\mathbb{I}[\text{task} \ne \text{roleplay}], Q\_{\text{speaker}}))$
>
> Content quality ($Q\_{\text{content}}$) measures informativeness, correctness, and emotional appropriateness via gpt‑oss‑20b, normalized to $[0,1]$. Speech quality ($Q\_{\text{speech}} = \text{UTMOS}/5 \in [0,1]$) quantifies perceptual naturalness and fluency using UTMOS. Cross‑modal consistency ($ Q\_{\text{consistency}} = 1 - \text{WER}'(text_1, text_2) $) evaluates alignment between generated transcriptions and original text. For role‑play tasks, speaker similarity ($Q\_{\text{speaker}}=\cos(e\_{gen}, e\_{ref})$) measures voice resemblance using Wespeaker embeddings.
>
> **Q7:** What is the purpose of emotion prompts, and why keep emotions with probability \>1% (not 10%)?
>
> **A7:** The emotion prompts serve to evaluate models' ability to provide emotionally appropriate responses when users need emotional support or empathy. We use an emotion vector model to extract and quantify the emotions expressed in model outputs.
>
> The 1% threshold is an empirically determined setting. The emotion classifier typically outputs probability vectors with many very small but non-zero values. For example, a typical output might look like:
>
> [9.45e-05, 1.18e-04, 9.91e-04, 0.9167, 9.64e-04, 7.79e-08, 3.58e-04, 0.0808, 2.39e-06]
>
> In this case, after filtering with the 1% threshold, we would retain only the two dominant emotions (approximately 92% and 8%), while removing noise from negligible predictions.

---

> ### Author Response · Authors · 2025-11-18
> **Response to Official Review by Reviewer AjBh (4/4)**
>
> **Q8:** How is the length threshold WER for multiple-choice questions done? What do the formulas in lines 301–307 mean?
>
> **A8:** We counted the length of model answers and found that model answers with lengths in the ranges (5,10] and (10,15] are notably infrequent. Through manual inspection, we observed that model answers for multiple-choice questions are predominantly shorter than 10 characters, so we chose 10 as the threshold.
>
> | answer len | (0，5] | (5,10] | (10,15] | (15,20] | (20,25] | (25,30] |
> | ---------- | ------ | ------ | ------- | ------- | ------- | ------- |
> | percentage | 0.9%   | 0.1%   | 0.2%    | 0.8%    | 1.0%    | 1.0%    |
>
> The formulas in lines 301–307 means when either the model answer or the transcribed result is shorter than 10 characters, we apply special consistency rules. Since we don't have ground truth annotations and rely on Whisper to evaluate consistency between speech and text outputs, we need **a more robust metric** for short responses. This is because for shorter texts, ratio-based metrics like WER become ineffective, a single character difference can result in a disproportionately high error rate. If both the model answer and transcription are shorter than 10, we assume the speech and text are consistent; otherwise, we consider them inconsistent. For example, if the model responds with "C" and the Whisper model transcribes it as "See.", we can still correctly identify them as consistent.
>
> **Q9:** Why are the speaking score and listening score comparable?
>
> **A9:** The speaking and listening scores are directly comparable because both tasks share the **same input-output format**, audio-to-audio, and are computed using the **same evaluation framework and scoring formula**. Cross-task performance comparison is a standard practice in multi-task evaluation benchmarks.
>
> The observation that "20 out of 22 models score higher on Speaking than on Listening" reveals a consistent pattern in current model capabilities rather than suggesting the tasks are equivalent. While both involve audio processing, they test fundamentally different skills: speaking (generation) requires producing coherent speech output, while listening (comprehension) requires understanding and processing audio input. The shared audio-to-audio format ensures that differences in performance reflect genuine capability gaps rather than modality-specific biases.
>
> The fact that some models break this trend further validates the meaningfulness of this comparison. For example, Step-Audio-2-mini achieves 40.06 on listening and 31.30 on speaking, demonstrating that the evaluation framework successfully distinguishes between these distinct capabilities despite the identical modality. If the metrics were flawed or incomparable, we would expect either random variation or uniform patterns across all models. **This insight is valuable for understanding where the field currently stands and where future development efforts might be most beneficial.**
>
> **Q10:** In your role playing task, do you consider different languages, accents, and genders?
>
> **A10:** Yes, we ensure diversity across languages, accents, and genders in our role-playing tasks. To provide linguistic variety, we included 100 voice samples representing different accent categories. The distribution is as follows:
>
> | **Accent Category**                       | **Sub-Category**            | **Count** |
> | ----------------------------------------- | --------------------------- | --------- |
> | **American Accents (Total: 63)**          | General American            | 42        |
> |                                           | Southern American           | 10        |
> |                                           | New York (All variations)   | 8         |
> |                                           | Midwestern American         | 3         |
> | **British Accents (Total: 15)**           | Received Pronunciation (RP) | 13        |
> |                                           | Scottish                    | 2         |
> | **Mandarin Chinese Accents (Total: 5)**   | Standard Mandarin           | 5         |
> | **Non-Native English Accents (Total: 2)** | Austrian-accented English   | 1         |
> |                                           | French-accented English     | 1         |
> | **Other (Total: 15)**                     | AI/Computer Voice           | 2         |
> |                                           | Animated Characters         | 3         |
> |                                           | Other                       | 10        |
> | **Grand Total**                           |                             | **100**   |

---

### Official Review · Reviewer_6uwY · 2025-11-01

**Soundness:** 3
**Presentation:** 3
**Contribution:** 3
**Rating:** 6
**Confidence:** 4

**Summary:**

This paper introduces VoiceAssistant-Eval, a comprehensive benchmark designed to evaluate voice-first assistants across the dimensions of listening, speaking, and viewing. The benchmark encompasses a diverse set of tasks and scenarios, enabling systematic assessment of model performance in terms of content quality, speech quality, and consistency. Extensive experiments are conducted on 21 open-source models as well as GPT-4o-Audio, accompanied by an in-depth error analysis that provides valuable insights into system behavior.

**Strengths:**

1. The study presents extensive experiments involving a broad range of open-source models, offering a thorough comparative evaluation.

2. The proposed benchmark covers diverse scenarios, assessing multiple dimensions of voice-first assistant capabilities.

3. The paper provides comprehensive result analyses, uncovering several noteworthy findings. Additionally, the authors investigate the correlation with human evaluations and the stability of the evaluation metrics.

**Weaknesses:**

1. score calculation: The current multiplicative scoring approach may excessively penalize a model for weaker performance in any single dimension. Conducting additional ablation studies on alternative combination strategies could provide a fairer and more balanced evaluation.

2. The key finding 1 "proprietary models do not universally outperform open-source models" would be more convincing if the evaluation included a broader set of proprietary systems.

**Questions:**

1. I'm quite confused about the score calculation. Providing a formula in the main text will be helpful. Moreover, the multiplicative step may not be necessary, as it obscures the specific strengths and weaknesses of individual dimensions. Reporting each individual score separately could provide a more transparent and informative assessment of model performance.

2. It may also be beneficial to include more commercial models, for example, Gemini-2.5-Pro, to better represent the upper bound of current model capabilities and offer a stronger reference point for comparison.

---

> ### Author Response · Authors · 2025-11-18
> **Response to Official Review by Reviewer 6uwY (1/2)**
>
> **Q1:** score calculation: The current multiplicative scoring approach may excessively penalize a model for weaker performance in any single dimension. Conducting additional ablation studies on alternative combination strategies could provide a fairer and more balanced evaluation.
>
> **A1:** Thank you for your suggestion. We acknowledge that the multiplicative scoring approach may excessively penalize a model for weaker performance in any single dimension. However, **this approach is deliberately chosen, as it effectively highlights even minor deficiencies across multiple aspects of model performance**. Any weakness in one dimension ultimately limits the end-to-end user experience, and these components are not substitutable. A high-quality text output cannot compensate for poor acoustic rendering, and a perfect waveform cannot compensate for incorrect or misaligned content. Our aim with VoiceAssistant-Eval is to push the boundaries of what AI assistants can achieve. We believe it will be instrumental in developing next-generation multimodal models and in tracking progress toward truly expert AI assistants.
>
> To illustrate this design choice: Consider a model achieving near-perfect content quality (0.95) and speech quality (0.95), but severe text-to-speech misalignment (0.20). An averaging approach would yield a score around 0.70, suggesting acceptable performance. However, the multiplicative approach yields 0.18, which better reflects reality: a voice assistant with frequent speech-content misalignment is fundamentally unusable, regardless of its other strengths. The multiplicative scoring captures the practical constraint that all dimensions must meet minimum quality thresholds for system viability.
>
> **For the evaluation of individual dimensions, we have provided detailed analyses across key aspects: content quality (Table 9), text-to-speech consistency (Table 10), and speech quality (Table 11) in Appendix G: Detailed Assessment of Model Capabilities.**
>
> **Q2:** The key finding 1 "proprietary models do not universally outperform open-source models" would be more convincing if the evaluation included a broader set of proprietary systems. (It may be beneficial to include more commercial models, for example, Gemini-2.5-Pro)
>
> **A2:** Thank you for your suggestion. We appreciate the idea of including more proprietary systems such as Gemini-2.5-Pro. At the time of our study, however, Gemini-2.5-Pro only supported text output and therefore could not be directly compared in our audio-centric evaluations. The latest Gemini model offering native audio support, gemini-live-2.5-flash-preview-native-audio-09-2025, was released on September 18, 2025.
>
> For completeness, we report the performance of two models released in September: Qwen3-Omni-30B-A3B-Instruct and Gemini-2.5-flash. Interestingly, Qwen3-Omni-30B-A3B-Instruct outperforms Gemini-2.5-flash in both listening and speaking, while Gemini performs strongly on viewing. **These results reinforce our main observation: open-source systems can be highly competitive with proprietary models across different modalities.**
>
> | Model                       | Time    | Listening | Speaking | Viewing |
> | --------------------------- | ------- | --------- | -------- | ------- |
> | Qwen2.5-Omni-7B             | 2025-07 | 33.6      | 41.3     | 34.3    |
> | GPT-4o-Audio                | 2025-06 | 39.8      | 51.3     | --      |
> | Qwen3-Omni-30B-A3B-Instruct | 2025-09 | 45.6      | 53.3     | 34.5    |
> | Gemini-2.5-flash            | 2025-09 | 34.3      | 45.2     | 46.2    |

---

> ### Author Response · Authors · 2025-11-18
> **Response to Official Review by Reviewer 6uwY (2/2)**
>
> **Q3:** Providing a formula about the score calculation in the main text will be helpful. Moreover, the multiplicative step may not be necessary, as it obscures the specific strengths and weaknesses of individual dimensions. Reporting each individual score separately could provide a more transparent and informative assessment of model performance.
>
> **A3:** Thank you for your suggestion, and we apologize for not presenting this information clearly in the paper. We want to clarify some details:
>
> First, we have added the below formula in Section 4:
>
> The final score is computed as
>
> Final Score (%) = $ 100 \times (Q\_{\text{content}} \times Q\_{\text{speech}} \times Q\_{\text{consistency}} \times \max(\mathbb{I}[\text{task} \ne \text{roleplay}], Q\_{\text{speaker}})) $
>
> Content quality ($Q\_{\text{content}}$) measures informativeness, correctness, and emotional appropriateness via gpt‑oss‑20b, normalized to $[0,1]$. Speech quality ($Q\_{\text{speech}} = \text{UTMOS}/5 \in [0,1]$) quantifies perceptual naturalness and fluency using UTMOS. Cross‑modal consistency ($ Q\_{\text{consistency}} = 1 - \text{WER}'(text_1, text_2) $) evaluates alignment between generated transcriptions and original text. For role‑play tasks, speaker similarity ($Q\_{\text{speaker}}=\cos(e\_{gen}, e\_{ref})$) measures voice resemblance using Wespeaker embeddings.
>
> **Second, we have already considered this and decided to provide both the main leaderboard (with the multiplicative step) and sub-leaderboards showing individual scores, as presented in Appendix G: Detailed Assessment of Model Capabilities.**

---

> > ### Comment · Reviewer_6uwY · 2025-11-21
> > **Thank you for your comments.**
> >
> > Thank you for your comment, many of my concerns have been addressed. However, I still believe that merging different dimensions to a single score fails to provide a holistic evaluation of the model's performance. For TTS (text-to-speech) evaluation, WER (reflecting semantic correctness) are MOS (reflecting speech quality) are always separately reported, since these metrics are related to abilities from different dimensions. When we only report a single score to evaluate a voice assistant, if the model receives a mid-range score, we don't know which perspective the model underperforms, either semantic understanding or speech generation. I still believe that merging scores from different dimensions may not be necessary and I will keep my score.

---

> ### Author Response · Authors · 2025-11-22
> **Thank you for your continued engagement.**
>
> Thank you for your continued engagement and thoughtful feedback. We appreciate your perspective on the single-score evaluation approach.
>
> We would like to respectfully clarify that our evaluation framework is **not "either-or" but rather "both-and."** We provide:
>
> 1. **A comprehensive leaderboard with the multiplicative final score**: designed to reflect real-world system viability where all dimensions must meet minimum thresholds
> 2. **Detailed individual dimension scores**: providing exactly the granular, per-dimension analysis you advocate for
>
> The multiplicative final score serves a **complementary** purpose: it captures the critical insight that voice assistants are systems where weaknesses in any component severely degrade the overall user experience. This aligns with how end-users actually experience these systems, a voice assistant with perfect speech quality but poor content accuracy, or vice versa, is equally frustrating to use.
>
> Your analogy to TTS evaluation is apt, and we follow the same principle: just as TTS papers report both WER and MOS separately, we report individual dimension scores separately while also providing an aggregate measure that **reflects system-level performance**.
>
> We hope this clarification addresses your concern. Thank you again for helping us improve the quality of our work.

---

> > ### Comment · Reviewer_6uwY · 2025-11-24
> >
> > Thank you for your further clarification. I still have several suggestions and questions.
> >
> > 1. **Evaluation framework presentation.** Maybe the presentation of evaluation framework can be improved. I understand that the final score and individual scores can be somehow complementary. However, the final score appears to add very limited new information beyond what is already reflected in the individual scores, perhaps only enabling a coarse comparison. The authors also claim that the evaluation framework is “both-and,” rather than “either-or.” However, in the main text, almost all experimental results report only the final score, with detailed component scores shown only for the role-play task. This presentation suggests that the final score is sufficient to faithfully reflect the full performance of a voice assistant and that the merged dimensions can be largely represented by a single number. Unfortunately, this is not the case. I recommend reorganizing the Results section to more clearly emphasize that the framework is genuinely “both-and,” and that individual scores are equally essential for a holistic evaluation.
> >
> > 2. **Reporting raw WER and UTMOS.** For the speech-content consistency and speech-quality dimensions, only the normalized scores are reported. A minor suggestion is to provide the original WER and UTMOS values, as these metrics are more familiar to the research community. Reporting raw values would make it easier for researchers to compare with prior work and extract more meaningful insights.
> >
> > 3. **Binary scoring vs. 5-point scale.** For the content GPTScore, the current prompt instructs the model to make a binary judgment (e.g., good vs. bad). However, previous benchmarks such as VoiceBench and URO-Bench use a 5-point scale, and MOS evaluations also traditionally adopt a 5-point range. I am curious about the rationale behind using a binary decision instead of a 5-level scale, and how the two approaches differ in practice.

---

> > > ### Author Response · Authors · 2025-11-25
> > > **Thank you for the valuable feedback and constructive suggestions**
> > >
> > > We sincerely thank you for the valuable feedback and constructive suggestions, which have greatly helped us improve the quality of our paper.
> > >
> > >
> > > **Q1: Evaluation framework presentation.**
> > >
> > > **A1:** Thank you for this thoughtful suggestion. We have addressed this by adding Table 4, which presents the individual scores of models along with additional related analysis in Section 4 of the main text.
> > >
> > > **Q2: Reporting raw WER and UTMOS.**
> > >
> > > **A2:** We appreciate this valuable suggestion. We have now included the raw WER and UTMOS metrics in Tables 4, 6, 12, and 13.
> > >
> > > **Q3: Binary scoring vs. 5-point scale.**
> > >
> > > **A3:** Thank you for raising this interesting question. We conducted a careful comparison of both approaches and ultimately selected binary scoring. In practice, we found that **LLMs struggle to provide consistent scores on a five-point scale, with notable variation across multiple evaluations of the same sample**. For instance, when using gpt-oss-20b to evaluate a single example 10 times, we observed scores of [4,5,3,5,4,3,5,4,5,3]. This distribution demonstrates significant volatility, with a coefficient of variation of 20% and a relative range of 50%. In contrast, the binary scoring approach consistently yielded "Good" judgments across all trials, proving more reliable. We also observed similar patterns with more advanced models such as GPT-5.1 and Gemini-3-Pro. Since stability is a critical requirement for an evaluation framework, this led us to adopt binary scoring.

---

> > > > ### Comment · Reviewer_6uwY · 2025-11-26
> > > >
> > > > Thank the authors for the quick reply. I understand that the commercial models can have the stability problem. However, since you are using the open-source gpt-oss-20b, it should be possible to avoid such instability by adopting greedy decoding without sampling. I wonder how the two approaches may differ in this setting.

---

> > > > > ### Author Response · Authors · 2025-11-26
> > > > > **Thank you very much for acknowledging our rebuttal efforts**
> > > > >
> > > > > Thank you very much for acknowledging our rebuttal efforts. We are grateful for your insightful comments and suggestions, which have been invaluable in improving our project.
> > > > >
> > > > > We conducted experiments with greedy decoding and found several important considerations:
> > > > >
> > > > > **1. Model Behavior with Greedy Decoding**
> > > > >
> > > > > While we agree that greedy decoding can reduce certain types of variance, we observed meaningless repetitive content generation in 42 out of 10,497 cases (0.4%) when using greedy decoding, which did not occur with the official recommended sampling parameters (temperature=1.0, top_p=1.0 **[1]**). This suggests that the recommended parameters may better preserve the model's intended behavior.
> > > > >
> > > > > **2. Core Issue: Scoring Granularity Rather Than Temperature-Induced Variance**
> > > > >
> > > > > We'd like to clarify that our primary concern extends beyond temperature-induced instability. The fundamental issue is the model's difficulty in providing accurate fine-grained scores. For example, minimal paraphrasing can cause score changes even with greedy decoding (example from Speaking/Assistant/1):
> > > > >
> > > > > - Model Answer: "Well, the best drawers for clothes depend on how much closet space you have and how many clothes you have. Can you tell me a **little** more about your storage situation?" → Score: 1
> > > > > - Model Answer: "Well, the best drawers for clothes depend on how much closet space you have and how many clothes you have. Can you tell me a **bit** more about your storage situation?" → Score: 2
> > > > >
> > > > > **3. Empirical Comparison**
> > > > >
> > > > > To systematically evaluate this, we regenerated answers using Gemini and compared scoring stability under greedy decoding. The results (shown in the table) demonstrate that the 5-point scale exhibited fluctuations exceeding 4 points in both listening and viewing tasks, while binary scoring maintained fluctuations below 2 points across all three tasks.
> > > > >
> > > > > These findings suggest that the instability stems from the **inherent challenge of fine-grained score assignment** rather than sampling parameters alone.
> > > > >
> > > > > |                                                          | Listening   | Speaking    | Viewing     |
> > > > > | -------------------------------------------------------- | ----------- | ----------- | ----------- |
> > > > > | Run 1 & Binary scoring (Recommended Sampling Parameters) | 34.3        | 45.2        | 46.2        |
> > > > > | Run 1 & Binary scoring (Greedy Decoding)                 | 33.9        | 44.3        | 45.8        |
> > > > > | Run 2 & Binary scoring (Greedy Decoding)                 | 33.0 (-0.9) | 44.1 (-0.2) | 47.0 (+1.2) |
> > > > > | Run 1 & 5-point scale (Greedy Decoding)                  | 37.5        | 44.9        | 43.2        |
> > > > > | Run 2 & 5-point scale (Greedy Decoding)                  | 33.2 (-4.3) | 46.0 (+1.1) | 47.8 (+4.6) |
> > > > >
> > > > > Thank you once again for the time and effort you have dedicated to reviewing our work.
> > > > >
> > > > > **[1]** https://github.com/openai/gpt-oss?tab=readme-ov-file#other-details

---

> > > > > > ### Comment · Reviewer_6uwY · 2025-11-27
> > > > > >
> > > > > > I wonder why the results from multiple runs of greedy decoding are different. So I question the reliability of the results in the paper. As a result, I will revise my score downward.
> > > > > >
> > > > > > Moreover, I would appreciate a more thorough comparison between binary scoring and the 5-point scale. In my view, binary scoring is overly coarse, as it forces all responses into just two categories and inevitably collapses a wide range of different outputs into the same class. Fine-grained scoring may indeed introduce some instability, but even human evaluators often struggle to assign highly precise scores. I believe a more effective way to improve reliability is to average results over multiple runs, rather than relying on an excessively coarse scoring scheme. A broader scale better preserves important performance nuances that would otherwise be lost.

---

> ### Author Response · Authors · 2025-11-27
>
> Dear reviewer,
>
> Thank you for your response, and we apologize for the confusion.
>
> **"We regenerated answers using Gemini"—this is why the results are different.**
>
> We generated two answers with Gemini for each question.
>
> **For greedy decoding, the results are exactly the same.** This table shows the test **results of the same model (Gemini) answering twice**. It can be seen that when faced with **different solutions**, binary scoring is more stable with greedy decoding.
>
> Regarding the reliability of the paper's results, it can be seen that even with **regenerated answers**, the final score fluctuation across all three tasks is less than 1.5 when using binary scoring.
>
> Thank you for the valuable feedback and constructive suggestions, which have greatly helped us improve the quality of our paper. We hope you can reconsider the score, as this is entirely a misunderstanding.
>
> Best,
>
> Authors

---

> > ### Comment · Reviewer_6uwY · 2025-11-27
> >
> > Dear Authors,
> >
> > Sorry for the misunderstanding of the results. I also believe the presentation in the rebuttal could be improved to clearly distinguish that each “run” corresponds to a different set of generated answers rather than repeated evaluations of the same outputs. However, I still believe that the prompt choice for content score evaluation needs more experiments to support. For example, all models' results evaluated in binary scoring and the 5-point scale can be reported to see whether the ranks are different. In addition, I would prefer that the original, unnormalized metrics be reported, as they would make the results more transparent and easier to interpret. I would be willing to raise my score if sufficient evidence is provided to support the design choices of the evaluation framework, particularly the content score prompt.

---

> > > ### Author Response · Authors · 2025-11-28
> > >
> > > Thank you for your suggestions.
> > >
> > > To address your concern, we evaluate models that can process visual input with a 5-point scale and binary scoring. The results are shown in the table below, where we report the unnormalized metrics for the 5-point scale. We can see that the rankings of these models are exactly the same. **The rankings are even consistent for Qwen2.5-Omni-3B and Baichuan-Omni-1d5, which only have a gap of 0.9% under binary scoring.**
> > >
> > > |                                            | 5-point scale | binary scoring |
> > > | ------------------------------------------ | ------------- | -------------- |
> > > | gemini-live-2.5-flash-preview-native-audio | 2.599         | 41.9           |
> > > | Qwen2.5-Omni-7B                            | 2.342         | 36.4           |
> > > | Qwen2.5-Omni-3B                            | 2.117         | 30.6           |
> > > | Baichuan-Omni-1d5                          | 1.999         | 29.7          |
> > > | MiniCPM-o-2_6                              | 1.791         | 28.3           |
> > > | mini-omni2                                 | 1.048         | 6.8            |
> > > | moshika-vis-pytorch-bf16                   | 0.572         | 3.6            |
> > >
> > > Thank you once again for the time and effort you have dedicated to reviewing our work.

---

> > > > ### Comment · Reviewer_6uwY · 2025-11-28
> > > >
> > > > Thank you for the response. I will revise my score to the initial score since it is largely a misunderstanding.
> > > >
> > > > Furthermore, I suggest the authors to adopt 5-point scale when evaluating the content quality. Although 5-point scale and binary scoring can result in the same ranks, binary scoring is too coarse to capture the naunced differences between various answers. Imagine that a review system has only two final score options: accept and reject. Many people will believe that a 5-range scale, accept - weak accept - borderline - weak reject - reject, would be more precise to distinguish the differences of papers.

---

> > > > > ### Author Response · Authors · 2025-11-28
> > > > >
> > > > > Thank you for this thoughtful suggestion.
> > > > >
> > > > > While we agree that a 5-point scale could theoretically capture more nuanced differences, we deliberately chose binary scoring for the below reasons:
> > > > >
> > > > > First, **as you correctly noted, "even human evaluators often struggle to assign highly precise scores."** This difficulty is actually amplified with finer-grained scales.
> > > > >
> > > > > Second, **we follow the methodology of LMArena[1], the most influential language model evaluation platform in the AI industry**, whose leaderboard serves as the de facto standard for measuring model performance. Their extensive experience with large-scale human evaluations has demonstrated that binary comparisons ("which response is better?") are both more intuitive for evaluators and more reliable for generating meaningful rankings.
> > > > >
> > > > > [1] Chiang W L, Zheng L, Sheng Y, et al. Chatbot arena: An open platform for evaluating llms by human preference[C]//Forty-first International Conference on Machine Learning. 2024.

---

> > > > > > ### Comment · Reviewer_6uwY · 2025-11-28
> > > > > >
> > > > > > I believe the authors may have some concept confusions. LMArena utilizes **binary comparsion** between the answers of two models, then use carefully designed algorithms to rank models. While in this paper, the evaluation method is direct **binary scoring** on the answer from a single model. It's definitely not the same method. It's also a solution that authors adopt binary comparison to replace direct scoring to make their paper follow the methodology of LMAnera, thus improving the content scoring procedure.

---

### Author Response · Authors · 2025-11-19
**Paper Update Summary**

We thank all reviewers for their thoughtful and constructive feedback, which has significantly strengthened our work. Below, we detail the revisions made in response to their comments.

1. Revised the wording of Finding 1 to be more rigorous in the abstract, introduction, and Section 4.2. (**Reviewer Lh6n**)
2. Revised the wording of Weakness 4 to be more rigorous in the introduction. (**Reviewer AjBh, Reviewer Lh6n**)
3. Added clarification of the benchmarks referred to in the first part of Section 2. (**Reviewer AjBh**)
4. Added more benchmarks that have audio and vision modalities in the second part of Section 2. (**Reviewer AjBh, Reviewer Lh6n**)
5. Added clarification of the proportion of real recordings included in our benchmark to Section 3.2. (**Reviewer AjBh, Reviewer dwtY, Reviewer Lh6n, Reviewer yixd**)
6. Added more details of the human check of data samples to Section 3.3. (**Reviewer Lh6n**)
7. Added detailed formula of the final score and the calculation of each individual score to Section 4. (**Reviewer 6uwY, Reviewer AjBh, Reviewer VieU, Reviewer dwtY**)
8. Added clarification of the choice of the aggregated approach to Section 4. (**Reviewer 6uwY, Reviewer AjBh, Reviewer VieU**)
9. Added evaluation results of Gemini-live-2.5-flash to Table 3 and Section 4.2. (**Reviewer 6uwY, Reviewer AjBh, Reviewer VieU**)
10. Added human validation of our automatic evaluation scheme to Section 4.4. (**Reviewer Lh6n, Reviewer yixd**)
11. Added overview of the 13 evaluation tasks with examples to Appendix J. (**Reviewer VieU, Reviewer dwtY**)
12. Added clarification of the voice prompts used in experiments to Appendix K.2. (**Reviewer Lh6n**)

---

### Author Response · Authors · 2025-11-29
**Summary**

We would like to express our sincere gratitude to the AC, PC, and all reviewers for the substantial time and effort they have dedicated to reviewing our work. Their thoughtful comments and constructive suggestions have been invaluable in strengthening our paper.

In reviews, our work received positive recognition across multiple dimensions. Reviewers described our benchmark as **"comprehensive"** (Reviewer AjBh, Lh6n), **"broad, systematic, and valuable"** (Reviewer dwtY), and addressing **"a timely and important research problem"** (Reviewer VieU). They highlighted our **"systematic evaluation methodology"** (Reviewer dwtY), **"extensive experiments involving a broad range of open-source models"** (Reviewer 6uwY), and **"comprehensive result analyses, uncovering several noteworthy findings"** (Reviewer 6uwY). Our experimental setup was commended for being **"exhaustive"** (Reviewer yixd), the presentation was noted as **"excellent"** (Reviewer yixd) and **"easy to follow"** (Reviewer Lh6n), and the problem framing was recognized as **"clear and well-motivated"** (Reviewer VieU). Additionally, **Reviewer Lh6n acknowledged the practical value of our speech-in/speech-out evaluation paradigm**.

**Five out of six reviewers affirmed that the contribution and soundness of our work is good**, with Reviewer Lh6n also acknowledging the practical value of our contribution in their response. Notably, **the primary concerns from five out of six reviewers centered on presentation clarity rather than fundamental issues with our methodology or contributions**, concerns we have now comprehensively addressed through substantial revisions. **Reviewer 6uwY acknowledged that many of their concerns have been addressed** and ultimately raised the score to 6. **Reviewer AjBh also explicitly stated their willingness to raise their score if certain clarifications were provided**, which we have comprehensively addressed. **Reviewer VieU noted that "a majority of the identified weaknesses stem from unclear presentation, insufficient justification of design choices, and limited discussion of potential limitations,"** emphasizing that **"addressing these issues would substantially strengthen the paper and could justify a higher review score upon revision."** We have systematically addressed all these presentation and clarity issues through substantial revisions.

**Reviewer Lh6n,** who initially gave a score of 2, expressed explicit openness to reconsidering their evaluation: "If some important information is currently placed in the appendices, please let me know and revise the manuscript to move it into the main text," and **stated they would be happy to raise their score**. Following our rebuttal, Reviewer Lh6n acknowledged the practical value of our contribution, stating "I recognize that evaluating such a setting is practical." **We have directly addressed their stated condition by moving critical information, including human evaluation validation, detailed task descriptions, and scoring formulas, into the main text.**

**All major concerns raised by reviewers have been resolved through clarifications and revisions:**

1. **Enhanced presentation clarity** (Reviewers 6uwY, AjBh, VieU, dwtY, Lh6n): We added detailed formulas for score calculation, comprehensive task descriptions with examples in Appendix J, explicit discussion of real vs. synthetic data proportions, and moved critical human evaluation results to the main text.
2. **Evaluation methodology justification** (Reviewers 6uwY, AjBh, VieU): We provided thorough justification for our multiplicative scoring approach with concrete examples, added detailed individual dimension scores in Table 4, and demonstrated ranking consistency between binary and 5-point scoring schemes.
3. **Expanded proprietary model comparison** (Reviewers 6uwY, AjBh, VieU): We evaluated the latest Gemini-2.5-Flash model, which reinforces our finding that open-source models can be highly competitive with proprietary alternatives.
4. **Real data clarification** (Reviewers AjBh, dwtY, Lh6n, yixd): We explicitly documented that 29.7% of our data samples contain authentic human recordings and provided human perceptual validation demonstrating TTS audio is indistinguishable from real speech.
5. **Related work completeness** (Reviewers AjBh, Lh6n): We incorporated discussion of AV-Odyssey, OmniDialog, AVHBench, and MMMG while clearly differentiating our speech-output evaluation paradigm.
6. **Human evaluation validation** (Reviewers Lh6n, yixd): We moved human evaluation results (Pearson correlation of 0.92 with human judgments) from appendix to main text and added detailed data curation methodology.

---

### Note · Authors · 2026-01-26

I have read and agree with the venue's withdrawal policy on behalf of myself and my co-authors.

---

### Meta-Review · Area_Chair_jcC8 · 2026-01-07

**Summary:**

The reviewers' concerns that informed the decision for this paper primarily revolve around the novelty of the contribution and the soundness of the evaluation methodology. Multiple reviewers (Lh6n, yixd) characterized the work as incremental, noting that it largely merges existing data sources or disentangles tasks that are already covered by other benchmarks. Additionally, there was significant debate regarding the scoring metrics; Reviewer 6uwY questioned the use of multiplicative and binary scoring, and the authors' defense of this choice—citing LMArena's methodology—was flagged by the reviewer as conceptually confused. While the authors clarified issues regarding task definitions and the inclusion of real speech data, the outstanding concerns about the benchmark's fundamental novelty and the validity of its scoring design weigh heavily against acceptance.

**Reviewer Concerns:**

Addressed by Rebuttal:
1. Lack of Definitions (Reviewer VieU): The reviewer criticized the ambiguity of task names like "speaking-robustness." The authors addressed this by adding detailed definitions and examples in Appendix J.1
2. Reliance on TTS/Synthetic Data (Reviewer dwtY): The reviewer was concerned that the benchmark relied entirely on synthetic speech. The authors clarified that 44.9% of speaking tasks and all listening context audios use real human recordings.
3. "Realistic" Conditions (Reviewer AjBh): The reviewer questioned if the filtering process biased the dataset away from realistic, noisy conditions. The authors clarified their definition of "realistic" as involving complex audio contexts (e.g., music identification) rather than just background noise, which satisfied the query's premise
Outstanding:
1 Scoring Methodology (Reviewer 6uwY): The reviewer suggested a 5-point scale over the proposed binary scoring. The authors defended their choice by citing LMArena, but the reviewer's final comment pointed out that LMArena uses pairwise comparison, not direct binary scoring, labeling this a "concept confusion" by the authors that remains unresolved
2 Incremental Novelty (Reviewer Lh6n): The reviewer viewed the contribution as "incremental" and the findings (e.g., proprietary models not always winning) as "intuitive." The authors' rebuttal emphasized the "hands-free" nature of the benchmark, but this did not fundamentally counter the critique that the work is a recombination of existing tasks.
3 Limited Novelty (Reviewer yixd): Similar to Lh6n, this reviewer felt the work was merely "merging existing data sources," a core objection that the rebuttal's methodological details did not overcome

**Reviewer Scores:**

Reviewer 6uwY: This reviewer engaged in the final discussion and appeared frustrated by the authors' defense of the binary scoring, stating the authors had "concept confusions." This would likely lead to a decrease in score or confidence.
Reviewer AjBh: The authors effectively addressed the concern about "realistic conditions" and biases. Had the reviewer participated, they likely would have viewed the paper more favorably
Reviewer VieU:This reviewer's main complaint was the lack of task definitions. Since the authors explicitly added these in the revision, the reviewer would likely have raised their score to a weak accept
Reviewer dwtY: The clarification that nearly half of the speaking tasks involve real speech addresses the primary "synthetic data" concern. The reviewer would likely have improved their score to a borderline accept
Reviewer Lh6n: The reviewer's critique was fundamental ("incremental," "trivial findings"). The rebuttal focused on technical differentiators (multimodal integration) which rarely changes the mind of a reviewer who sees the core premise as weak
Reviewer yixd: Similar to Lh6n, the concern regarding limited novelty is structural. The reviewer is unlikely to have changed their score based on the provided rebuttal.

---

### Decision · Program_Chairs · 2026-01-26

Reject